# A Survey of Utilization and Satisfaction of Korean Subfertility Treatment among Korean Women

**DOI:** 10.3390/healthcare12161600

**Published:** 2024-08-12

**Authors:** Minjung Park, Seungwon Shin, Jihye Kim, Jong-hyun Kim, Dong-Il Kim, Soo-Hyun Sung, Jang-Kyung Park

**Affiliations:** 1Department of Public Health and Administration, Seoul Digital University, Seoul 07654, Republic of Korea; 2College of Korean Medicine, Sangji University, Wonju 26339, Republic of Korea; ssw.kmd@gmail.com; 3Research Institute of Korean Medicine Policy, The Association of Korean Medicine, Seoul 07525, Republic of Korea; 4Department of Medical Classics and History, College of Korean Medicine, Gachon University, Seongnam 13306, Republic of Korea; 5Department of Obstetrics and Gynecology, College of Korean Medicine, Dongguk University, Gyeongju 38066, Republic of Korea; 6Department of Policy Development, National Institute for Korean Medicine Development, Seoul 04516, Republic of Korea; 7Department of Korean Medicine Obstetrics and Gynecology, Pusan National University Korean Medicine Hospital, Yangsan 50612, Republic of Korea

**Keywords:** infertility, subfertility, Korean medicine, survey

## Abstract

Low fertility is a critical social problem worldwide, and infertility has a prevalence of 15%. This cross-sectional study aimed to understand the factors affecting the usage and satisfaction of Korean medicine (KM) in subfertile women. An online survey was conducted from 3 November to 8 November 2021. The survey collected basic information, KM treatment experience, and satisfaction from women who experienced poor pregnancy. The *t*-test and chi-square test (χ^2^-test) were used to determine the overall characteristics of the subjects and factors affecting the utilization and satisfaction of KM treatment. Of the total of 29,465 people, 4922 read the survey email, and 601 responded. After excluding 51 respondents with questionable response patterns, 550 respondents were included in the final analysis. Of these, 43.1% (*n* = 237) had experience with conventional treatment, and 16.5% (*n* = 91) had received KM treatment. The group that received both KM treatment and CM treatment (*n* = 59, 24.9%) was significantly more prevalent than the group that received KM treatment alone (*n* = 32, 10.2%) (*p* = 0.00). Women who had given birth more than once or held a master’s degree were significantly more willing to participate in the ‘KM Support Project for Subfertility’ program. Our findings suggest that subfertile patients prefer integrated treatment that combines KM and CM treatments. Further studies are needed to assess the status of integrative medicine treatment, satisfaction with each KM intervention, factors for low satisfaction, and patient requirements.

## 1. Introduction

Infertility is a significant and growing concern worldwide, affecting millions of couples and individuals who struggle to conceive [1]. It is defined as the failure to establish a clinical pregnancy after 1 year of regular unprotected sexual intercourse or due to a fertility impairment of the individual or their partner [1]. This condition not only brings emotional and psychological stress but also impacts physical health and social relationships [2].

Infertility is a common condition affecting approximately 10–15% of the global population, with its age-standardized prevalence rate on the rise [1]. This rise can be attributed to various factors including delayed childbearing, lifestyle changes, and environmental influences [3]. As a potential public health issue, infertility can cause disability, lead to relationship problems, and threaten community well-being [4]. Additionally, about 13.5% of Korean women of childbearing age suffer from infertility [5],with many experiencing mental health issues such as depression [2].

The introduction of assisted reproductive technology (ART) has increased pregnancy success rate in patients with infertility, leading to a rise in global demand for ART [6]. Since 2006, the Korean government has provided medical support for ART to address the country’s ultra-low birth rate [7]. However, in vitro fertilization (IVF), a common ART procedure, has adverse effects, including ovarian hyperstimulation syndrome, multiple pregnancies, severe maternal morbidity, and increased preterm birth, cerebral palsy, and infant mortality [8,9]. Addressing the unmet emotional and physical needs of patients during ART is crucial [6,10], and the high cost of IVF poses a societal burden [11].

In response to these challenges, the demand for Korean medicine (KM) has been steadily increasing for treating infertility, either alone or as an adjuvant to ART. KM helps regulate hormones to promote reproductive health [12,13] and is used globally for infertility treatment [13,14]. In Korea’s dualized medical system, KM is widely used for infertility treatment. Despite the lack of central government support [15], local governmental projects supporting infertile couples with KM have shown high satisfaction, and this increases the demand for central governmental support [16,17,18].

However, previous surveys on satisfaction and demand were conducted among participants who were already familiar with Korean medicine, which may have biased the results to be more favorable towards KM. Furthermore, since there were no surveys targeting the general population, there is lack of research on the reasons and decision factors for choosing KM among women with subfertility, as well as their satisfaction with it.

Therefore, this study aims to investigate the correlation between the experience and satisfaction of using KM and the intention to use KM in the future among women with poor pregnancy outcomes and to inform future policy development. We conducted a survey exclusively targeting women who have difficulty conceiving, focusing on obtaining relatively accurate information about the subfertile women population. The survey provides comprehensive information, including trends and utilization of subfertility treatments, factors influencing their use, satisfaction levels, and willingness to participate in a future government-supported KM subfertility treatment program.

## 2. Materials and Methods

### 2.1. Survey Design and Study Sample

The data for this study were collected through a survey on Korean women’s perceptions of subfertility. The survey participants were women experiencing subfertility, defined as those who had not become pregnant despite having regular sexual intercourse for over 1 year if they were under 35 years or for 6 months if they were 35 years or older and were registered with an online survey company. The survey was conducted over 6 days, from 3 November to 8 November 2021, with emails sent to 29,465 individuals. Of these, 4922 accessed the emails, and 601 responded. Among the nonresponders, 4251 were ineligible, and 70 individuals discontinued their responses midway. Out of the 601 respondents, 550 were included in the final analysis, after excluding 51 responses that showed insufficient effort that were answered in straight lining (i.e., answering a series of questions in the same way, like marking all answers in a straight line down a column) [19]. The study sample utilized panel data held by the research organization Woori Bio. The selection process of the participants is shown in Figure 1.

This study did not involve direct manipulation of humans for research purposes, did not focus on specific research subjects initially, and did not collect sensitive information. Consequently, it was approved for exemption from review by the Institutional Review Board (IRB) of Pusan National University Korean Medicine Hospital (PNUKHIRB 2021-10-016).

### 2.2. Development of the Survey Form

Based on the results of previous studies [15], a draft questionnaire was developed and evaluated for a facial validity by 10 women from the general population. Subsequently, two KM professors specializing in obstetrics and gynecology reviewed and revised the questionnaire. The survey method used was a self-reported online panel survey, consisting of two main parts: the first part gathered information about the respondents, while the second part contained questions related to conventional medicine and KM treatment. The questions about respondent information included sex, age, experience in planning pregnancy and subfertility, marital status, pregnancy experience, educational level, height, weight, average monthly household income, occupation, and area of residence. The questions related to KM treatment covered experiences and satisfaction with KM treatment and willingness to participate in the central government’s Korean Medicine Support Project for Subfertility. The conventional medicine treatments mentioned included ovulation induction (administration of clomiphene citrate), ART (in vitro fertilization or intrauterine insemination), and surgical treatment. KM treatment included herbal medicine, acupuncture, moxibustion, pharmocopuncture, cupping, and Chuna manual therapy.

### 2.3. Statistical Analyses

For the descriptive analysis, the *t*-test and chi-square test (χ^2^-test) were used to understand the overall characteristics of the participants and to identify the factors affecting the utilization of KM treatment and its satisfaction. The *t*-test was chosen because it is effective for comparing the means of two groups to determine if there is a statistically significant difference between them. The chi-square test was selected because it is suitable for assessing relationships between categorical variables and determining if there are significant associations between participant characteristics and their use of KM treatment. For a more detailed analysis adjusting for potential confounders, logistic regression analysis was performed to evaluate the factors influencing the intention to participate in the KM subfertility support program. Logistic regression was chosen because it is appropriate for modeling binary outcome variables and allows for the adjustment of multiple confounding factors, providing a more accurate assessment of the factors influencing the intention to participate. The data were analyzed using IBM SPSS Statistics 21 (IBM Corp., Armonk, NY, USA) and R version 4.2.1, with the significance level set at 5%.

## 3. Results

### 3.1. Characteristics of Respondents According to Their Experience of KM Treatment

The results of the cross-analysis between the demographic factors and KM are presented in Table 1. Of the 550 women who participated in the survey, 83.5% (*n* = 459) had no experience with KM treatment, while 16.5% (*n* = 91) had experience. Among those who had never received conventional treatment, 10.2% (*n* = 32) had experience with KM treatment. Conversely, among those who had experience with conventional treatment, 24.9% (*n* = 59) also had experience with KM treatment. There were significantly more individuals who had experience with subfertility treatment in both conventional medicine (CM) and KM (*p* = 0.00).

### 3.2. Characteristics of Respondents According to Satisfaction with KM Treatment

A subgroup analysis was conducted targeting only respondents who reported having experience with KM treatment to identify factors affecting satisfaction. The 91 participants were classified into a high satisfaction group (*n* = 36, 39.6%) and a low satisfaction group (*n* = 55, 60.4%) (Table 2). The high satisfaction group was defined as those who were “satisfied with the KM treatment” when the response was “high (*n* = 32)” or “very high (*n* = 4)”. The low satisfaction group was defined as those who were “insufficiently satisfied with the KM treatment” when the response was “normal (*n* = 40)”, “low (*n* = 14)”, or “very low (*n* = 1)”.

The proportion of women with high satisfaction was significantly higher among those who were pregnant but had not yet given birth, as well as those who had more than one childbirth experience, compared to women who had never been pregnant (*p* = 0.00).

Regarding the willingness to participate in the Korean Medicine Support Project for Subfertility, the intention to participate was significantly higher in the group with high satisfaction with KM treatment (*p* = 0.01).

The concurrent use of other CM treatments and the diagnosis of subfertility had no significant effect on satisfaction with KM treatment.

### 3.3. Factors Affecting Intention to Participate in KM Subfertility Support Program

Table 3 shows the factors influencing the intention to participate in the KM subfertility support program, based on logistic regression analysis of all survey participants. Through this analysis, it will be possible to identify population groups likely to accept the expansion of the subfertility support program to KM in the future. The result indicates that having a graduate school education (master’s degree) significantly increases the willingness to participate in the KM subfertility support program (Coef. ± S.E. = 1.09 ± 0.41, odds ratio = 2.99, Z value = 2.63, *p* = 0.01). Conversely, women with more than one childbirth experience showed significantly lower willingness to participate in the program compared to women with no pregnancy experience (Coef. ± S.E. = −0.78 ± 0.28, odds ratio = 0.46, Z value = −2.80, *p* = 0.01).

## 4. Discussion

In this study, 16.5% of the 550 respondents had experience with subfertility therapies in KM. Among the respondents, 10.2% of women who had no experience with subfertility treatment in conventional medicine clinics visited KM clinics. Approximately 24.9% of subfertile female patients undergoing treatment in conventional medicine opted for cotreatment in KM clinics, a statistically significant difference (*p* = 0.00).

This discrepancy might be because KM therapies could alleviate the undesirable symptoms that occur during conventional medical treatments. For instance, one study proposed that acupuncture is effective in controlling anxiety in subfertile women [20]. Moreover, the results reflect the expectation of subfertile women that the coadministration of integrative medicine treatments might be more effective than a singular approach. Several studies support this expectation, suggesting that combination treatment with KM treatment could be an encouraging way to treat female subfertility [21,22,23]. In this study, age, household income, education level, region, and job type were found to have no significant impact on satisfaction with KM treatment. Several thoughts might explain these results. First, KM treatment has traditionally been accessible to people of all ages and social backgrounds. Second, KM treatment provides personalized treatments based on the individual’s condition and adopts a holistic approach that emphasizes overall health, leading to uniform satisfaction and expectations among diverse backgrounds. Third, the sample of participants in this study may not have been biased towards any particular group, resulting in an even distribution of individuals from various backgrounds. However, this study is a retrospective survey, and the possibility of recall bias cannot be excluded. Additionally, the reliance on self-reported data limits the accuracy of the responses. The study’s findings may have been influenced by other demographic factors not considered, and the sample size limitation requires caution in generalizing the results.

The proportion of women who reported being satisfied was significantly higher among those who had experienced pregnancy and childbirth compared to those who had never been pregnant (*p* = 0.01). This could be interpreted as possibly due to achieving pregnancy through KM treatment. Women with various reproductive problems experience showing significant satisfaction with integrative medicine, which is eco-friendly and reduces side effects. KM can improve the mental health of infertile female patients undergoing treatment by encouraging normal sexual activities [20]. KM is a holistic approach to managing overall health, including reproductive health [11]. Therefore, many infertile women seek integrative medicine services. Traditionally, KM is widely used for infertility treatment in Korea, and many infertile couples recognize that KM is effective and safe for infertility treatment [12,24]. A nationally conducted survey in Korea showed that 58.3–63.3% of infertile females underwent several KM therapies before receiving ART [25]. Since KM has been used for a long time, there is a perception that it is safe and effective, leading women to seek KM when they encounter pregnancy problems. This positive perception is thought to have influenced the high level of satisfaction among those who succeeded in pregnancy and childbirth through KM treatment. Several additional factors might be considered to explain the high satisfaction rates among women who experienced pregnancy and childbirth through KM treatment: Achieving pregnancy itself can provide significant psychological relief and a sense of accomplishment for women who have struggled with infertility, contributing to higher satisfaction rates [26]. Pregnancy and childbirth can improve the overall quality of life and well-being, leading to higher satisfaction levels [27]. The holistic approach of KM may lead to improvements in other health conditions, which can positively impact overall satisfaction [28].

The intention to participate in the Korean medicine support project for subfertility was generally high, regardless of the participants’ satisfaction levels, and the group with high satisfaction with KM treatment showed a significantly higher intention to participate in the program than the unsatisfied group (*p* = 0.01).

Lastly, the results of a logistic regression analysis of factors affecting the willingness to participate in ‘KM Subfertility Support Program’ for all women showed that women with a master’s degree or higher and women with no pregnancy experience had significantly higher willingness to participate. Therefore, in the future, it can be expected that the participation of women with no experience of pregnancy or childbirth will be high. This also suggests that the expansion of subfertility programs in the future may be highly acceptable to highly educated women, especially women who have no experience of pregnancy. It appears that women with a master’s degree or higher are expected to be older compared to other groups (college graduates or below), which is believed to have influenced their intention to participate in the program. As age increases, fertility declines, making it a major factor in increasing subfertility rates [29]. However, it was found that there was almost no difference in the intention to participate in the program between the group aged 34 and younger and the group aged 35 and older (*p* = 0.98), suggesting the need for additional research.

There may be additional factors influencing the intention to participate in the KM subfertility support program. First, financial incentives can influence the intention to participate in medical support programs. Offering subsidies or financial assistance for KM treatments can make these programs more attractive to a broader range of women, particularly those who might otherwise be deterred by the cost. Second, cultural attitudes towards KM and subfertility treatments play a vital role in shaping women’s intentions to participate. In Korea, where KM is traditionally respected and widely accepted, positive cultural perceptions can enhance participation rates. Third, the influence of social networks and peer support can encourage women to participate in the KM support project. Positive experiences and endorsements from friends, family, and community members who have benefitted from the program can motivate others to join.

Based on the results of this study, additional research is needed to identify the relationship between pregnancy, childbirth, satisfaction with KM treatment, and participation in the ‘KM Subfertility Support Program.’ It is possible that women who successfully conceived and gave birth through KM treatment reported high satisfaction due to their success. Conversely, those who never participated in the KM support project for subfertility might have responded without full awareness of the government’s cost support or might have been influenced by word of mouth. Furthermore, the participants were not necessarily diagnosed with subfertility but were those who reported difficulty in becoming pregnant. This led to a smaller number of participants undergoing subfertility treatment compared to previous studies, indicating the need for further research to reflect the current status of KM.

The limitations of this study were as follows. First, the study sample does not represent the entire population, as the data were sourced from a private company’s panel. Therefore, there is a limitation that the results of the study cannot be generalized. In South Korea, a national survey on the usage of Korean medicine, known as the National Survey for Usage and Consumption of Korean Medicine (National Approved Statistics Number: 117087), is conducted every two years [30]. In the future, it is necessary to incorporate the survey items used in this study into the national survey to ensure that investigations are conducted on a national sample. Second, because the survey relies on respondents’ self-reported experiences, there is potential for recall bias and inaccuracies in the results. Therefore, survey research requires careful interpretation of the results. Third, due to the short survey period of 5 days, only 601 out of 29,465 individuals were included in the study. In future survey research, it is necessary to have a sufficient survey period of at least 10 days to increase the response rate. Finally, while facial validity of the draft questionnaire was assessed, content validity and reliability were not evaluated. Reliability and validity are crucial procedures for ensuring the credibility of research results.

To confirm our findings and explore areas not covered due to the limitations of our study, future research should include the following: an in-depth study on the current status of conventional and KM combination treatments; the effects of each KM intervention, side effects, satisfaction levels, and factors contributing to low satisfaction; the development of a more respondent-friendly questionnaire, as a detailed survey on satisfaction and the demand for various KM treatment interventions was not conducted; further investigation into financial incentives, cultural attitudes, accessibility, and social influences that might impact the intention to participate in KM support programs; and an in-depth analysis of the psychological impact of KM treatments on subfertility patients. This includes studying how psychological factors such as stress, anxiety, and depression affect treatment outcomes. These suggestions will underline the ongoing relevance of our work and encourage additional exploration in this field. By addressing these areas in future research, we can provide a clearer and more balanced analysis of the utilization and satisfaction of Korean subfertility treatments.

## 5. Conclusions

In conclusion, this survey showed that patients with subfertility are more likely to receive KM as part of integrative medical treatment rather than KM as a standalone intervention. Our findings also suggest that experience with pregnancy and childbirth has a statistically significant effect on satisfaction with KM treatment compared to those without pregnancy experience. The results of this study support the recommendation to implement the ‘KM Subfertility Support Program’ when establishing policies to address low birth rates.

## Figures and Tables

**Figure 1 healthcare-12-01600-f001:**
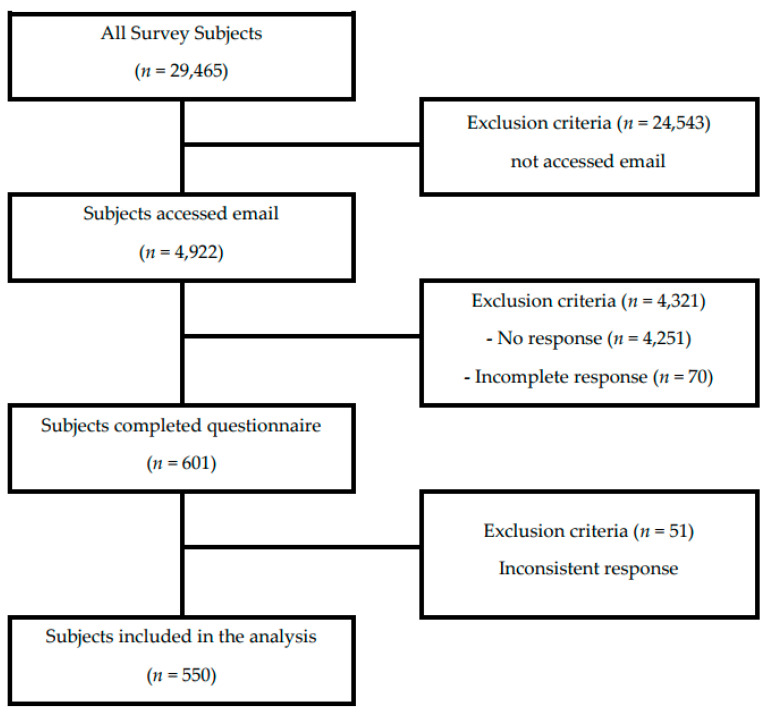
Flow chart of selected study population.

**Table 1 healthcare-12-01600-t001:** Demographic and clinical characteristics of participants.

		KM txt = 0 (*n* = 459, 83.5%)	KM txt = 1 (*n* = 91, 16.5%)	Total (*n* = 550)	
Age	Under 34	98 (79.7%)	25 (20.3%)	123 (22.4%)	*p* = 0.80
	35 and more	361 (84.5%)	66 (15.5%)	427 (77.6%)	
Household income	Under 4000 USD	132 (81%)	31 (19%)	163(29.6%)	*p* = 0.90
	Over 4000 USD	327 (84.5%)	60 (15.5%)	387 (70.4%)	
Education	Under high school	77 (86.5%)	12 (13.5%)	89 (16.2%)	*p* = 0.76
	University (bachelor’s degree)	343 (83.9%)	66 (16.1%)	409 (74.4%)	
	Graduate school (master’s degree) or higher	39 (75%)	13 (25%)	52 (9.5%)	
Region	Rural area	203 (83.2%)	41 (16.8%)	244 (44.4%)	*p* = 0.96
	Urban area	116 (86.6%)	18 (13.4%)	134 (24.4%)	
	Capital	140 (81.4%)	32 (18.6%)	172 (31.3%)	
Job	Manual	32 (76.2%)	10 (23.8%)	42 (7.6%)	*p* = 0.67
	Non-manual	224 (81.8%)	50 (18.2%)	274 (49.8)	
	Others	203 (86.8%)	31 (13.2%)	234 (42.5%)	
Obesity	BMI < 25	372 (82.3%)	80 (17.7%)	452 (82.2%)	*p* = 0.65
	BMI ≥ 25	87 (88.8%)	11 (11.2%)	98 (17.8%)	
Delivery history	Never been pregnant	91 (85.8%)	15 (14.2%)	106 (19.3%)	*p* = 0.92
	Pregnant but never gave birth	47 (73.4%)	17 (26.6%)	64 (11.6%)	
	More than 1 childbirth experience	321 (84.5%)	59 (15.5%)	380 (69.1%)	
Infertility diagnosis	No	319 (85.5%)	54 (14.5%)	373 (67.8%)	*p* = 0.46
	Yes	140 (79.1%)	37 (20.9%)	177 (32.3%)	
	Cause of subfertility (*n* = 177)				*p* = 0.99
	Unexplained	75 (81.5%)	17 (18.5%)	92 (29.0%)	
	Tubal or peritoneal factor	19 (82.6%)	4 (17.4%)	23 (7.3%)	
	Ovulatory factor	25 (71.4%)	10 (28.6%)	35 (11.0%)	
	Male factor	14 (77.8%)	4 (22.2%)	18 (5.7%)	
	Others	7 (77.8%)	2 (22.2%)	9(2.8%)	
CM txt	No	281 (89.8%)	32 (10.2%)	313 (56.9%)	*p* = 0.00 **
	Yes	178 (75.1%)	59 (24.9%)	237 (43.1%)	
Intention to participate in KM Support Project for Subfertility	No	216 (86.4%)	34 (13.6%)	250 (45.5%)	*p* = 0.11
	Yes	234 (81.0%)	57 (19.0%)	300 (54.5%)	

Signif. codes: 0.001 ‘**’. CM: conventional medicine, KM: Korean medicine, txt: treatment, USD: United States dollar, urban area: metropolitan city or state, rural area: small city or county, job others: housewives, students, part-time jobs, and unemployed, cause of subfertility others: diminished ovarian reserve, thin endometrium, and leiomyoma of uterus.

**Table 2 healthcare-12-01600-t002:** A cross-analysis between the demographics and satisfaction of KM treatment.

		Unsatisfied (*n* = 55, 60.4%)	Satisfied (*n* = 36, 39.6%)	Total (*n* = 91)	
Age	Under 34	40 (60.6%)	26 (39.4%)	66 (72.5%)	*p* = 1.00
	35 and more	15 (60.0%)	10 (40.4%)	25 (27.5%)	
Household income	Under 4000 USD	20 (64.5%)	11 (13%)	31 (34.1%)	*p* = 0.73
	Over 4000 USD	35 (58.3%)	25 (41.7%)	60 (65.9%)	
Education	Under high school	6 (50.0%)	6 (50.0%)	12 (13.2%)	*p* = 0.62
	University (bachelor’s degree)	40 (60.6%)	26 (39.4%)	66 (72.5%)	
	Graduate school (master’s degree) or higher	9 (69.2%)	4 (30.8%)	13 (14.3%)	
Region	Rural area	23 (56.1%)	18 (43.9%)	41 (45.1%)	*p* = 0.71
	Urban area	12 (66.7%)	6 (33.3%)	18 (19.8%)	
	Capital	20 (62.5%)	12 (37.5%)	32 (35.2%)	
Job	Manual	5 (50.0%)	5 (50.0%)	10 (11.0%)	*p* = 0.77
	Non-manual	31 (62.0%)	19 (38.0%)	50 (54.9%)	
	Others	19 (61.3%)	12 (38.7%)	31 (34.1%)	
Obesity	BMI < 25	49 (61.3%)	31 (38.8%)	80 (87.9%)	*p* = 0.75
	BMI ≥ 25	6 (54.5%)	5 (45.5%)	11 (12.1%)	
Delivery history	Never been pregnant	15 (100%)	0 (0%)	15 (16.5%)	*p* = 0.00 **
	Pregnant but never gave birth	10 (58.8%)	7 (41.2%)	17 (18.7%)	
	More than 1 childbirth experience	30 (50.8%)	29 (49.2%)	59 (64.8%)	
Infertility diagnosis	No	32 (59.3%)	22 (40.7%)	54 (59.3%)	*p* = 0.95
	Yes	23 (62.2%)	14 (37.8%)	37 (40.7%)	
	Cause of subfertility (*n* = 37)				*p* = 0.94
	Unexplained	10 (58.8%)	7 (41.2%)	17 (45.9%)	
	Tubal or peritoneal factor	3 (75.0%)	1 (25.0%)	4 (10.8%)	
	Ovulatory factor	7 (70.0%)	3 (30.0%)	10 (27.0%)	
	Male factor	2 (50.0%)	2 (50.0%)	4 (10.8%)	
	Others	1 (50.0%)	1 (50.0%)	2 (5.4%)	
CM txt	No	17 (53.1%)	15 (46.9%)	32 (35.7%)	*p* = 0.41
	Yes	38 (64.4%)	21 (35.6%)	59 (64.8%)	
Intention to participate in KM Support Project for Subfertility	No	27 (79.4%)	7 (20.6%)	34 (37.4%)	*p* = 0.01 *
	Yes	28 (49.1%)	29 (50.9%)	57 (62.6%)	

Signif. codes: 0.001 ‘**’, 0.01 ‘*’. CM: conventional medicine, KM: Korean medicine, txt: treatment, USD: United States dollar, urban area: metropolitan city or state, rural area: small city or county, job others: housewives, students, part-time jobs, and unemployed, cause of subfertility others: diminished ovarian reserve, thin endometrium, and leiomyoma of uterus.

**Table 3 healthcare-12-01600-t003:** Factors affecting intention to participate in KM subfertility support program.

			Coef. ± S.E.	Odds Ratio	Z Value	Pr (>|z|)
Age	(ref. over 35)					
	Under 34		0.62 ± 0.26	1.87	2.37	*p* = 0.98
Household income	(ref. under 4000 USD)					
	Over 4000 USD		−0.04 ± 0.21	0.96	−0.20	*p* = 0.84
Education	(ref. high school)					
	University (bachelor’s degree)		0.15 ± 0.25	1.17	0.61	*p* = 0.54
	Graduate school (master’s degree) or higher		1.09 ± 0.41	2.99	2.63	*p* = 0.01 *
Region	(ref. Rural area)					
	Urban area		0.28 ± 0.23	1.32	1.20	*p* = 0.23
	Capital		−0.05 ± 0.22	0.95	−0.26	*p* = 0.79
Job	(ref. Manual)					
	Non-manual		0.48 ± 0.36	1.61	1.39	*p* = 0.16
	Others		0.14 ± 0.24	1.15	0.42	*p* = 0.68
Obesity	(ref. BMI < 24)					
	BMI ≥ 25		0.16 ± 0.24	1.17	0.69	*p* = 0.49
Delivery history	(ref. Never been pregnant)					
	Pregnant but never gave birth		−0.41 ± 0.36	0.67	−1.11	*p* = 0.27
	More than 1 childbirth experience		−0.78 ± 0.28	0.46	−2.80	*p* = 0.01 *
diagnosis	(ref. No)					
	Yes	Unexplained	−0.00 ± 0.28	1.00	0.06	*p* = 0.95
	Tubal or peritoneal factor		−0.38 ± 0.48	0.69	−0.77	*p* = 0.44
	Ovulatory factor		0.10 ± 0.41	1.10	0.30	*p* = 0.77
	Male factor		0.19 ± 0.53	1.20	0.40	*p* = 0.69
	Others		−0.08 ± 0.72	0.92	−0.10	*p* = 0.91
CM txt	(ref. No)					
	Yes		−0.15 ± 0.46	0.86	0.58	*p* = 0.56
KM txt	(ref. No)					
	Yes		0.17 ± 0.32	1.18	1.18	*p* = 0.24
Intercepts			−0.03 ± 0.49	0.97	−0.02	*p* = 0.98
AIC = 746.64						

Signif. codes: 0.01 ‘*’. CM: Conventional medicine, KM: Korean medicine, txt: treatment, USD: United States dollar, urban area: metropolitan city or state, rural area: small city or county, job others: housewives, students, part-time jobs, and unemployed, cause of subfertility others: diminished ovarian reserve, thin endometrium, and leiomyoma of uterus.

## Data Availability

The raw data supporting the conclusions of this article will be made available by the authors on request.

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
