# Peer review of "A Survey of Utilization and Satisfaction of Korean Subfertility Treatment among Korean Women"

_healthcare, 2024, doi:10.3390/healthcare12161600_

Round 1
Reviewer 1 Report (New Reviewer)
Comments and Suggestions for Authors
Dear Authors,
Thank you for the opportunity to review your manuscript titled "A Survey of Utilization and Satisfaction of Korean Subfertility Treatment among Korean Women." Your work addresses a significant area in healthcare, and I commend your efforts in this important research. Here are a few suggestions and concerns that could help improve the clarity and impact of your findings, which are possible to address even after the completion of your study:
Feedback for Improvement:
-
Clarify the Sample Representation:
- Suggestion: Provide a more detailed description of your sample population to acknowledge any limitations in generalizability.
- Rationale: Explicitly discussing the characteristics of your survey sample and how they might differ from the general population can help readers better understand the context of your findings.
-
Discuss Limitations in Self-reported Data:
- Suggestion: Add a section discussing the limitations of relying on self-reported data and how this might affect your results.
- Rationale: Acknowledging the potential for recall bias and inaccuracies will provide a more balanced interpretation of your findings.
-
Address Short Survey Duration:
- Suggestion: Mention the short duration of the survey in the limitations section and discuss how it might have impacted the results.
- Rationale: This transparency will help readers understand the constraints under which the data were collected.
-
Exclusion of Insincere Responses:
- Suggestion: Elaborate on the criteria used to exclude insincere responses and how this exclusion might affect the results.
- Rationale: Providing details on the exclusion process will strengthen the validity of your data analysis.
-
Analyze and Interpret Demographic Factors:
- Suggestion: Expand the discussion on the influence of demographic factors such as region, job type, and educational level on KM treatment satisfaction.
- Rationale: A more in-depth analysis and discussion can provide valuable insights and address potential underestimations in the initial findings.
-
Contextualize High Satisfaction Rates:
- Suggestion: Discuss other potential factors contributing to the high satisfaction rates among those who experienced pregnancy and childbirth, beyond KM treatment alone.
- Rationale: This will help avoid overestimating the impact of KM treatment and provide a more nuanced interpretation of your results.
-
Address Motivational Factors for Participation:
- Suggestion: Reflect on other factors that might influence the intention to participate in the KM Support Project, such as financial incentives or cultural attitudes, in your discussion.
- Rationale: This will provide a more comprehensive understanding of the factors driving participation intentions.
-
Highlight the Importance of Further Research:
- Suggestion: Emphasize the need for further studies to confirm your findings and explore areas that were not covered due to the limitations of your study.
- Rationale: Suggesting areas for future research will underline the ongoing relevance of your work and encourage additional exploration in this field.
By addressing these suggestions, your manuscript will provide a clearer and more balanced analysis of the utilization and satisfaction of Korean subfertility treatments. These improvements can be made through revisions and additional discussions in your current manuscript. I appreciate your attention to these concerns and look forward to seeing the enhanced version of your work.
Best regards,
Comments on the Quality of English LanguageMinor editing:
Grammar and Syntax Issues:
Examples:
"The proportion of women with high satisfaction being significantly higher among those who were pregnant but had no childbirth experience and those who had more than one childbirth experience compared to women who had no pregnancy experience (P=0.00)." (awkward phrasing)
"Intention to participate in the Korean Medicine Support Project for Subfertility was generally high regardless of satisfaction." (incomplete structure and clarity)
Suggestions: Each sentence should be reviewed for proper grammar and syntax to ensure clarity and readability.
Clarity and Cohesion:
Examples:
"However, previous surveys on satisfaction and demand were conducted among project participants familiar with KM, which may result in outcomes favorable to KM." (unclear)
"Future research should include an in-depth study on the current status of conventional and KM combination treatments, the effects of each KM intervention, side effects, satisfaction levels, and factors contributing to low satisfaction." (run-on sentence)
Suggestions: Improve sentence structure to enhance clarity and ensure that each paragraph flows logically from one idea to the next.
Consistency in Terminology:
Examples:
The manuscript uses terms like "subfertility," "infertility," "KM treatment," and "conventional treatment" inconsistently.
Suggestions: Standardize terminology throughout the manuscript to avoid confusion.
Author Response
We appreciate the time and effort the reviewer has dedicated to reviewing this paper. Please see the attachment.

Reviewer 2 Report (New Reviewer)
Comments and Suggestions for Authors
Summary: The summary mentions percentages and P values ​​without any context, which can be confusing. Make sure that each percentage and statistical value is properly explained.
The statement “Our findings suggest that subfertile patients prefer integrative medical treatment over KM” needs to be explained for clarity.
Introduction: The introduction should provide a comprehensive background to the topic. It currently jumps into statistics without properly setting the context.
The statement “Approximately 13.5% of women of childbearing age in Korea face infertility” should be referenced immediately after this statistic.
There are inconsistencies in the formatting of the survey design and participant details. For example, the statement “Survey participants were women who had not become pregnant despite having regular sexual intercourse for a certain period of time (1 year for those under 35 and 6 months for those over 35)” should be formatted more clearly.
The description of the statistical tests used should be more detailed. Provide the rationale behind the selection of certain tests, such as the t-test and chi-square test.
The discussion section needs to be more in-depth. It currently reiterates some points made in the results without providing further analysis or context.
The conclusion should not provide new information. At this time, "More is needed to determine the status of KM and Western medical companion therapy, satisfaction levels for each KM intervention, and studies of factors contributing to low satisfaction and need" is more appropriate for the discussion section.
Comments on the Quality of English LanguageMinor editing
Author Response
We appreciate the time and effort the reviewer has dedicated to reviewing this paper. Please see the attachment.

Reviewer 3 Report (New Reviewer)
Comments and Suggestions for Authors
See attached

Author Response
We appreciate the time and effort the reviewer has dedicated to reviewing this paper. Please see the attachment.

Round 2
Reviewer 1 Report (New Reviewer)
Comments and Suggestions for Authors
Thank you so much for addressing the comments
This manuscript is a resubmission of an earlier submission. The following is a list of the peer review reports and author responses from that submission.
Round 1
Reviewer 1 Report
Comments and Suggestions for Authors
General comments:
The manuscript under review presents a summary of a survey performed to assess the satisfaction from the use of Korean Traditional Medicine among subfertile women. The collected data certainly has value per se. However, the manuscript text misses a clear statement as to why this survey has been performed. Some questions pop up. For instance, what were the anticipated outcomes of the authors prior to the start of the survey? Did the authors observe something un-intuitive? Is there any potential in translating any of these results into a patient care or additional education on the subject while undergoing fertility treatment? What is the overarching goal?
The current analysis boils down to two summary tables, however more sophisticated presentation of the data could be performed. For instance, rather than binning different responses into 2 groups, the authors could visualize data by plotting continuous variables (eg, age) against income and/or color-code by education level group, or other responses. Some regression models should be implemented and interpreted. Overall, more thorough analysis and modeling of the data is requested.
Please find below a list of specific comments in order as they appear in the manuscript.
Line 21: The authors use "infertile women" here and in many places below. A term “subfertile” is probably more suitable in the context of this study.
Line 24: The statement "responses of 550 women from a panel of 29,465 were analyzed" does not align with the text in the methods where you indicate that only ~4K read about the survey and only ~600 responded to the survey request. Please correct.
Line 27: The following sentence is wrongly written "The ratio of women who were pregnant but never gave birth to those who had given birth at least once was significantly higher than that of women who had never given birth (P=0.00)."
Line 45: "pregnancy success in patients with verifiable infertility," -- call it subfertility
Line 80: How were the "unfaithful respondents" determined?
Line 90: Is the average monthly household income calculated per person? Please clarify.
Line 108: The term "conventional infertility treatment" has not been clarified. Please add some information.
Line 112: The term "conventional medicine" has not been clarified. Please add some information.
Table 1: The currency used to represent income is missing.
Line 124: "The proportion of women who were pregnant but had no childbirth experience and those who had more than one childbirth experience were significantly greater than those who had never been pregnant". It is unclear, how is this related to "satisfaction"? Please clarify.
Line 127: Use 'the respondents' instead of “they”, otherwise, specify which subgroup you are referring to exactly.
Several entries in Table 2 need clarification:
- Education: Does "University" include BSc (3 years) and MSc (5 years)? If possible, divide by BSc (or equivalent), MSc (or equivalent) and PhD (or equivalent). In some countries gradutae school means MSc + PhD, in other these are two disjoint programs each awarding a title in a different discipline.
- Region: Rural/Urban: add mean density information instead?
- Job: replace "etc" with "other". Also, explain what other could be if it is not “manual” and not “not-manual”.
Additional analyses/discussion:
- Investigate and discuss how infertility diagnosis and delivery history could potentially confound satisfaction of TKM infertility treatment and how it could be reflected in the overall Intention to participate in Korean medicine Support Project for Subfertility by subfertile women with no prior experience (eg, discouraging statements made by word of mouth).
Author Response
We appreciate the time given and efforts made by the editor and referees in reviewing this paper. Please see the attachment.

Reviewer 2 Report
Comments and Suggestions for Authors
Please include some information About reproductive problem in your. Samples.
Please include name of infertility in different table
Please more explain about conventional treatment
Please Include limitation of this research.
Please declare strict your conclusion,
Author Response

(The authors gave the same response as above.)

Round 2
Reviewer 1 Report
Comments and Suggestions for Authors
General comments:
The use of "subfertility" in the first line in the abstract is incorrect: "Low fertility is a critical social problem worldwide, and subfertility is a common disease, with a prevalence of 15%."
The use of "subfertility" in the first line in the introduction is also incorrect: "Subfertility is defined as the failure to establish a clinical pregnancy after 1 year of regular unprotected sexual intercourse or owing to a fertility impairment of the individual or partne fertility"
In both cases, the authors mean infertility. "Subfertile" should be used in the context where patient was experiencing difficulties to establish pregnancy. Basically, infertile patients are infertile, whereas subfertile patients experience pregnancy but have a tough time getting pregnant and/or maintaining pregnancy. Infertile and subfertile should not be used interchangeably, but adequately to the context. Please research the two terminologies and use appropriately.
For that, whenever you cite an existing literature, make sure you use terminology from the cited literature (infertility/subfertility) or argue why different terminology was used otherwise.
In the discussion the authors note that it is "necessary to educate the general public about the methods and advantages of subfertility treatment using TKM". The authors should make more effort to articulate what are the advantages and what does TKM entails exactly.
Specific comments:
line 89: it is unclear what straight lining means
Table 2: the numbers do not add up for different categories, eg, age category has much less participants than education group. This should be noted and discussed.